# Current State of Hyperspectral Remote Sensing for Early Plant Disease Detection: A Review

**DOI:** 10.3390/s22030757

**Published:** 2022-01-19

**Authors:** Anton Terentev, Viktor Dolzhenko, Alexander Fedotov, Danila Eremenko

**Affiliations:** 1All-Russian Institute of Plant Protection, 3 Podbelsokogo Str., Pushkin, 196608 Saint Petersburg, Russia; vid@iczr.ru; 2World-Class Research Center «Advanced Digital Technologies», Peter the Great St. Petersburg Polytechnic University, 29 Polytechnicheskaya Str., 195251 Saint Petersburg, Russia; danilaeremenko@mail.ru

**Keywords:** remote sensing, hyperspectral, plant diseases, early detection, oil palm, citrus, cereals, solanaceae

## Abstract

The development of hyperspectral remote sensing equipment, in recent years, has provided plant protection professionals with a new mechanism for assessing the phytosanitary state of crops. Semantically rich data coming from hyperspectral sensors are a prerequisite for the timely and rational implementation of plant protection measures. This review presents modern advances in early plant disease detection based on hyperspectral remote sensing. The review identifies current gaps in the methodologies of experiments. A further direction for experimental methodological development is indicated. A comparative study of the existing results is performed and a systematic table of different plants’ disease detection by hyperspectral remote sensing is presented, including important wave bands and sensor model information.

## 1. Introduction

The spread of various, including invasive, plant diseases and pests is one of the most important problems in modern agriculture [1]. Therefore, to solve these relevant problems, the timely monitoring of plant diseases and pests is necessary. Remote sensing methods hold great promise for solving these problems [2]. Remote sensing data can identify crop conditions, including diseases, and provide useful information for specific agricultural management practices [3,4].

There are two types of remote sensing technologies: passive (such as optical) and active remote sensing (such as LiDAR and radar). Passive optical remote sensing is usually divided into two groups based on the spectral resolution of the sensors used: multispectral and hyperspectral remote sensing [5]. Hyperspectral sensing shows great potential as a non-invasive and non-destructive tool for monitoring biotic and abiotic plant stress among passive remote sensing methods, which measure reflected solar radiation [6]. This method collects and stores information from the spectroscopy of an object in a spectral cube that contains spatial information and hundreds of contiguous wavelengths in the third dimension. Hyperspectral imaging offers many opportunities for the early recognition of plant diseases by providing preliminary indicators through subtle changes in spectral reflectance due to absorption or reflection. Hyperspectral images with hundreds of spectral bands can provide detailed spectral portraits, hence, they are better able to detect subtle variations in soil, canopies or individual leaves. Thus, hyperspectral images can be used to solve a wider class of problems for the accurate and timely determination of the physiological status of agricultural crops. The early identification of disease spread and pest outbreaks may avoid not only significant crop loss, but also reduce pesticides usage and mitigate their negative impacts on human health and the environment, thus, improving the existing IPM [7,8].

In recent years, a wide range of miniature hyperspectral sensors available for commercial use have been developed, such as Micro- and Nano-Hyperspec (Headwall Photonics Inc., Boston, MA, USA), HySpex VNIR (HySpex, Skedsmo, Skjetten, Norway) and FireflEYE (Cubert GmbH, Ulm, Germany) [9]. These sensors can be installed on manned or unmanned airborne platforms (for example, airplanes, helicopters, and UAVs) to obtain hyperspectral imaging and support various monitoring missions [10,11].

There are various types of hyperspectral cameras, e.g., push-broom cameras, whisk-broom cameras and snapshot cameras. The measurement principle of each sensor type depends on its ability to obtain the whole picture (snapshot) at one time, one line of the picture (push broom) or one point of the picture (whisk broom) [12].

The general routine of collecting and processing hyperspectral images is presented in Figure 1. The light reflected from plant leaves is collected by the hyperspectral camera (Figure 1A) [13]. A hyperspectral data cube (Figure 1B) is obtained from the hyperspectral camera. Then various data normalization (Figure 1C) and feature extraction (Figure 1D) algorithms are applied to reduce the data’s dimensionality. Finally, different automatization techniques are used to automate the classification process (Figure 1E).

Hyperspectral remote sensing provides image data with very high spectral resolution [16,17]. This high resolution allows subtle differences in plant health to be recognized. Such a multidimensional data space, generated by hyperspectral sensors, has given rise to new approaches and methods for analyzing hyperspectral data [18,19].

For a long time, feature extraction methods have been used that reduce the data dimension without loss (or with minimal loss) of the original information on which the classification of hyperspectral images is based [20]. One of the most widely used dimensionality reduction techniques in HRS is principal component analysis (PCA). PCA computes orthogonal projections that maximize data variance and outputs the dataset in a new, uncorrelated coordinate system. Unfortunately, the informational content of hyperspectral images does not always coincide with such projections [21]. Thus, other methods are also used for feature extraction. The common methods for extracting hyperspectral data used in pathological research traditionally include PCA [22], derivative analysis [23], wavelet methods and correlation plots [24]. Alternatively, the hyperspectral image data can be processed at the image level to extract either spatial representation alone or joint spatial spectral information. If only spatial features are considered, for example, when studying structural and morphological features, spatial patterns among neighboring pixels with relation to the current pixel in the hyperspectral image will be extracted. Machine vision techniques, such as using a two-dimensional CNN, with a *p* × *p* chunk of input pixel data have been implemented to automatically generate high-level spatial structures. Extraction of spatial characteristics, in tandem with spectral elements, has been shown to significantly improve model performance. [25]. The use of spatial spectral characteristics can be achieved using two approaches: (i) by separately extracting spatial characteristics using CNN [26,27] and combining data from a spectral extractor using RNN, or LSTM [27,28]; and (ii) by using three-dimensional patterns in hyperspectral data cubes (*p* × *p* × b) associated with *p* × *p* spatially adjacent pixels and b spectral bands to take full advantage of important distinctive patterns.

In preparing this review, we tried to determine whether there is a general experimental method by which to achieve consistent results in the detection of plant diseases using hyperspectral remote sensing (HRS). We planned to identify existing gaps and tried to find solutions to level those gaps by analyzing existing publications. We believed that the main gaps could be related to the biological aspect of the experiments [29,30,31] and to the incorrect definition and interpretation of wavebands important for plant disease detection, which is also strongly related to biological aspects, namely plant physiology and biochemistry [31,32,33,34]. Considering the machine methods for analyzing hyperspectral data, we believe that, despite the advances in such techniques, such as ANN, SVM and others, their usage for identifying plant diseases with HRS is only a matter of choosing methods for data processing automation. Thus, in this review we will not discuss the advantages or disadvantages of different machine learning methods, especially since these issues have already been discussed by other authors in [35,36,37,38] and other papers.

There are many works devoted to the topic of plant disease detection using HRS; therefore, an urgent task is to prepare a review of hyperspectral remote sensing according to those articles whose authors tried to solve the problem of early detection of plant diseases as one of the key tasks for improving the existing IPM [39,40,41,42]. The early detection of plant diseases is, for a number of reasons, much more difficult than detecting them at the stage of visible symptoms. We believe that the knowledge of methods for identifying plant diseases at the symptomatic stage is the basis for their early detection at the asymptomatic stage. For this reason, we have included these articles in the review along with articles on the early detection of plant diseases using HRS.

The primary search for data on the topic of early detection of plant diseases was carried out using the following keywords (hyperspectral; plant diseases; plant pests; early; detection) during the period from 2006 to 2021. The most important data, selected on the basis of an analysis of the experience gained on the topic, are presented in the form of tables concluding each section of the review.

The choice of plant cultures mentioned in the review was dictated by the need for a sufficient sample of information for analysis. Thus, after analyzing the available articles, we opted for four crops: oil palm, citruses, *Solanaceae* family plants and wheat. A number of articles devoted to the early detection of diseases in various crops also will be mentioned but without detailed analysis because of lack of sufficient information. Though the low number of articles devoted to the crops different from oil palm, citruses, Solanaceae family plants and wheat made it impossible to perform a deep study or detect dependences in the successful or unsuccessful usage of HRS for plant disease early detection of other species. Thus, the objective of the article was set to analyze the current state of hyperspectral remote sensing for early plant disease detection of four different crop types: oil palm, citruses, *Solanaceae* family plants and wheat. In our opinion, the selection of these plant species represents a sufficiently representative sample to identify the main advantages and disadvantages of HRS in relation to the early plant diseases detection with generalization to other crops.

So, the main objective of our article was to prove the possibility of early plant disease detection by hyperspectral remote sensing. Another scientific assumption that authors tried to verify is that the spectral reflectance (i.e., important bands) should coincide (possibly with some small shift) with the same diseases and plants. Another objective of this review, then, was to systematize the modern research carried out in the field of using HRS for the detection—Primarily the early detection—Of plant diseases. Within this analysis, the available results are summarized and the main gaps in the field of early detection of plant diseases with HRS are highlighted.

The rest of this paper is organized as follows. Section 2 reviews the current state of hyperspectral remote sensing for early plant disease detection in four types of plants in detail (Section 2.1 for oil palm, Section 2.2 for citrus, Section 2.3 for the *Solanaceae* family, Section 2.4 for wheat). Due to a lack of information for comprehensive analysis, all other crops are jointly reviewed in Section 2.5. Section 2.6 is the summary for the reviewed materials. Section 3 discussed found gaps and problems, and conclusions are presented in Section 4.

## 2. Materials and Methods

### 2.1. Hyperspectral Remote Sensing of Oil Palm Diseases

The palm oil is used in many different ways and is a leader amongst other vegetable oils on the world market; this is why it is very important to control palm oil pests and diseases [43]. There are not so many of them; nevertheless they may adversely affect the palm oil harvest [44]. This section highlights the articles on the detection of such diseases using HRS.

The most important disease of oil palm is basal stem rot (BSR) caused by *Ganoderma boninense* [45,46]. This disease is a major threat to sustainable oil palm production that can reduce yields by 80% [47,48]. *Ganoderma boninense* are capable of degrading lignin to carbon dioxide and water, and then use the celluloses as nutrients for the fungus, destroying the host plant in the process [49].

Lelong et al. studied the possibility of discriminating several levels of *Ganoderma boninense* fungus contamination on oil palm trees’ canopy hyperspectral reflectance data. Using the PLS-DA method, a global performance accuracy of 92–98% was achieved [50,51]. Shafri et al. investigated the possibility of identifying oil palm diseases using HRS, as well as applying various vegetation indices to such data. It was possible to achieve accuracy over 80% for various indices, however, it was concluded that red, edge-based techniques are more effective than vegetation indices in detecting BSR-infected oil palm trees [52,53,54]. In later studies, the possibility of discriminating between three classes of BSR disease severity (healthy, mild and severe symptoms) was examined. A dataset of hyperspectral snapshots of various distances was used to discriminate BSR severity with variable degrees of success [55]. The work continued, in the article, on an optimal SVI development for the early detection of BSR in oil palm seedlings. The authors used the developments and the experience of earlier studies; the significant and insignificant wavelengths and indices were selected from [52,53,54,55]. The wavelengths then were used to create a SVI for the early detection of BSR. The result of the work was the determination of the best indices, presumably most suitable for the early detection of BSR in oil palm [56]. In their next article, the authors continued to improve the technology by performing a thorough analysis of airborne hyperspectral images using different SVIs, red edge position, and continuum removal [57].

Concerning the early detection of BSR, in 2014 Liaghat et al., for the first time, achieved this goal. The authors investigated the capability of reflectance spectroscopy to detect BSR at three stages of infection, and the results confirmed the applicability of VIS-NIR spectroscopy to classify BSR-infected oil palm leaves from healthy samples in early stages of the infection. The goal was the possibility of detecting infected oil palms in early stages, which was successfully achieved with an overall accuracy rate of 97% (without false-negatives) when a k-NN-based classification model was used [58]. Ahmadi et al. took into account not only the HRS, but also the weather data (temperature, precipitation and relative humidity). Two datasets were generated under different weather conditions, the dry and rainy season respectively. The data processing was carried out by using various ANNs. The plants were classified into four groups, healthy and mildly, moderately and severely infected ones. It was possible to obtain 100% classification accuracy for the mildly infected palms that were not showing any visible symptoms, thus, achieving the goal of the early detection of BSR. An important observation concerns the fact that ANNs showed better performance in identifying the diseased palms rather than healthy ones, with the remark that although the studied healthy palms did not have BSR, they may have suffered from other diseases or stresses that influenced their spectral properties [59]. Azmi et al. used different types of SVM to identify disease symptoms. It should be noted that the authors studied a very large number of articles on the topic and presented their data in the form of a comparative table. Unlike other studies where the experiments were carried out on an oil palm plantation or nursery, this experiment was carried out in a greenhouse with artificial, constant conditions. In this study, NIR reflectance showed significant differences between the inoculated *Ganoderma boninense* and healthy subjects. The infection could be detected early even in the absence of physical symptoms of the disease using SVM classifiers with different numbers of NIR bands. It was mentioned that using a large number of bands provided high classification accuracy, while a lesser number of bands gave slightly less accuracy. The authors supposed that the developed method needs to be tested in an open environment in order to confirm its reliability for field usage, taking into account the peculiarities of work with sunlight angle, shade and weather conditions [60].

The Orange Spotting (OS) is an another oil palm disease, caused by the coconut cadang-cadang viroid (CCCV), which has killed over 40 million coconut palms only in the Philipinnes [61,62]. Currently, for oil palms this disease appears to be of minor importance. However, due to its high severity on coconut and other palms, it is being studied to prevent oil palm epiphytotics, which CCCV may cause in the future due to possible mutations. Interesting results were obtained by Selvaraja et al. studying OS on oil palm trees. Using a HRS dataset for various SVI, the authors discovered that OS could be detected in symptomatic oil palm trees. In an earlier study the authors were able to determinate oil palm trees with potassium stress from those with OS, which can be useful in oil palm plantation management [63,64]. Another group of authors conducted a number of studies, including the publication of a review article on various ANNs for plant disease detection using HRS [36]. On the subject of OS, the authors conducted and published a number of studies on SVI and ANN choice for disease determination, as well as chlorophyll content at the leaf scale of the diseased plants. The hyperspectral data of OS diseased and healthy oil palm seedlings was processed by five different ANNs for evaluation of four red-edge indices followed by the selection of spectral bands from the red edge (680–780 nm), with a result that a red-edge inflection point (at 700 nm) could serve as a good indicator of the plant stress caused by OS in oil palm seedlings [65,66]. A systematization of the reviewed materials is present in Table 1.

### 2.2. Hyperspectral Remote Sensing of Citrus Diseases

The citrus fruits, which are grown in more than 75 countries around the world, are an important commercial crop. The most threatening diseases in the citrus industry are citrus bacterial canker (CBC), caused by *Xanthomonas citri* [67,68], and citrus greening disease, also known as Huanglongbing (HLB), primarily caused by a bacterium, *Candidatus liberibacter* spp., spread by insects [69]. These bacteria interfere with the nutrient supply of citrus trees until the infected trees die. HLB’s diagnosis methods are mainly based on genetic methodologies, such as PCR. An effective and environmentally friendly method of treatment has not yet been found, and the only measure to slow down or reduce further infestation is to destroy the affected trees [70,71,72].

Various authors have published a number of articles devoted to the determination of HLB using HRS. Sankaran et al. used portable halogen lamps mounted on a platform to provide an additional illumination to citrus leaves in outdoor studies of HLB [73,74]. After several studies the authors achieved an average overall classification accuracy of 87% with a minimum number of false negatives, using SVM to analyze the dataset of healthy and diseased samples [75]. The effect of wind and the presence of HLB-infected leaves within the canopy were mentioned as an additional factor, which resulted in some spectral variations [73]. Li et al. used red-edge position in the field and laboratory experiments on Valencia and Hamlin orange cultivars to achieve a noticeable difference between healthy and HLB-damaged canopies. The indoor dataset achieved higher accuracy than the outdoor dataset (about 95% vs. about 90%) due to a better environment and more samples. Overall, different spectral feature analyses of different datasets were achieved between 43 and 95% accuracy [76]. Kumar et al. obtained more accurate results when using a multispectral rather than a hyperspectral camera (87% vs. 80% accuracy). The MTMF method proved to be the most successful for hyperspectral images, as was the SAM method for multispectral ones [77]. Weng et al. described the classification models for healthy, HLB-infected (asymptomatic and symptomatic) and nutrient-deficient citrus leaves of the citrus Unshiu and Ponkan, which achieved accuracies of 90.2%, 96.0% and 92.6% for the cool season, the hot season and the entire period, respectively, using LS-SVM. The authors have demonstrated the possibility of hyperspectral reflection imaging combined with analysis of citrus carbohydrate metabolism for the detection of HLB in different seasons and cultivars. The classification model developed for the Satsuma cultivar dataset was successfully used for HLB detection of the Ponkan cultivar by calibration model transfer, and obtained an overall detection accuracy of 93.5% with a low rate of false negatives [78].

A greatest success in early the detection of HLB was achieved by Deng et al., with a study on early non-destructive detection and grading of citrus HLB disease [79,80,81,82,83]. The research was able to provide three models of early diagnosis and the grading of HLB disease by taking advantage of the PLS-DA method, tested with a leave-one-out cross-validation strategy. In the third model established with preprocessed spectral reflectance data by Savitzky-Golay, the smoothing and first-derivative methods had the best discrimination, which achieved a prediction accuracy of no less than 92% on five kinds of leaf samples, and an overall classification accuracy rate of 96.4%. In subsequent works the authors used multiple machine learning algorithms (logistic regression, decision tree, SVM, k-NN, LDA and ensemble learning) to distinguish between the groups of healthy and HLB-infected (symptomatic and asymptomatic) samples, based on the reflectivity of leaves. In the three-group classification (healthy and symptomatic/asymptomatic HLB leaves), SVM achieved an accuracy of 90.8%, while in two-group classification (healthy and symptomatic HLB leaves) accuracy reached 96%. The results showed that a small number of bands is not enough for stable classification; meanwhile, 13 characteristic bands identified by the proposed method provided the best performance. The team continued the study, researching the possibility of determining HLB using two different hyperspectral cameras installed on a UAV. The pixel-level-based HLB classification accuracy was 99.33% for the training set and 99.72% for the verification set.

There are also two interesting studies on the successful (96–100% accuracy) early detection of decay in citrus using HRS. The fruit damage discussed in these publications was caused by *Penicillium digitatum* fungy. Although this topic relates more to the post-harvest crop storage than the plant diseases, we saw fit to mention these articles [84,85]. The systematization of the reviewed materials is presented in Table 2.

### 2.3. Hyperspectral Remote Sensing of Solanaceae Plant Diseases

The *Solanaceae* family, which includes tomatoes, potatoes, tobacco, peppers and other crops, is one of the most common vegetable crops, both in greenhouses and outdoors. There are a number of diseases affecting these crops, such as early and late blight [86,87], different viruses [88,89], bacterial and target spot [90] and others, that can cause serious losses to yields. Due to the high economic harmfulness of these diseases, it is very important to detect them at early stages in order to apply timely and proper control measures. This section highlights articles on the detection of such diseases using HRS, including their early detection.

Successful studies determining the pathogenic states of tomato plants using hyperspectral sensing have been undertaken by a number of research teams from different countries. Lu et al. studied yellow leaf curl virus and late blight caused by *Phytophthora infestans*, target spot caused by *Corynespora cassicola* and bacterial spot caused by *Xanthomonas perforans* on tomato leaves in laboratory conditions, using multiple spectral vegetation indices selected by PCA, and reached up to 100% accuracy, including early-stage detection [91,92]. Polder et al. investigated the possibility of detecting potato virus Y (PVY) with the CNN method and achieved 75–92% accuracy [93]. Griffel et al. used PLS-DA and SVM classification method to achieve 89.9% accuracy in PVY detection [94]. Van De Vijver et al. studied early blight caused by *Alternaria solani* in the Bintje potatoe variety with spectral analysis and reached up to 92% accuracy [95]. Abdulridha et al. studied yellow leaf curl, target spot and bacterial spot in tomato leaves of the Charger and FL-47 cultivars in field and laboratory conditions, using different vegetation indexes, and obtained accuracies of 94–100% for determining different diseases from each other and 98–100% for determining healthy from diseased plants [96,97]. Zhang et al. investigated late blight on potatoes using HS image processing with MNF and SAM and successfully detected the disease with unmentioned accuracy [98]. Fernandez et al. studied late blight on Shepody cultivar potatoes with different spectral indices applied in the 400–900-nm diapason and achieved 85–91% accuracy [99], and used the same method in red and red-edge diapasons (660–780 nm) with 89% accuracy [100]. Krezhova et al. used methods of statistical and derivative analyses to detect tomato spotted wilt virus (TSWV) on the tobacco cultivar Samsun NN. At the asymptomatic stage the differences in reflectance spectra were statistically significant [101]. Xie et al. detected early and late blight [102] and gray mold [103] on tomato leaves of the Zheza 809 cultivar. They used an ELM classifier model for late and early blight and a k-NN model for gray mold, with the former achieving 94 to 100% accuracy, while between 44 and 66% accuracy was achieved for detecting asymptomatic gray mold diseased tomato leaves, 1 day past inoculation.

All of the authors managed to achieve a stable definition of the symptoms of the diseases in the visible stages. Most of the experiments were carried out in the field. The articles [96,97] should be noted separately, as the field experiments in these studies were duplicated by laboratory ones, as recommended in [29,30,31], and we suppose that it was why the authors achieved the best results in the accuracies of determining both various diseases and the differences between healthy and diseased plants.

Wang et al. presented two studies on the early detection of TSWV on sweet pepper with CNN analysis of HRS data. In the first study, the authors processed hyperspectral shots taken with a camera in the range of 400–1000 nm under laboratory conditions with a new GAN architecture, named as OR-AC-GAN. For the pixel-level classification, the false positive accuracy rate was 1.47% for healthy plants [104,105]. In second study the OR-AC-GAN was further improved and achieved a result in of 96.25% accuracy in the early detection of TSWV before visible symptoms showed up. The statistic results from the proposed OR-AC-GAN model were superior to the results of direct CNN model and AC-GAN model [105]. Gu et al. also successfully detected TSWV on tobacco in the early stages using three wavelength selection methods (SPA, BRT and GA), and four machine learning techniques (BRT, SVM, RF and CART). Among the selected bands, most were located at the NIR region (780–1000 nm). The models built by the BRT algorithm using the wavelengths selected by SPA obtained the best overall accuracy of 85.2% [106]. Zhu et al. studied tobacco mosaic virus (TMV) on tobacco of the MS Yunyan 87 cultivar. It was shown that it is possible to detect the TMV disease in the range of 450 to 1000-nm wavelengths with the usage of different machine learning algorithms, i.e., SVM, BPNN, ELM, LS-SVM, PLS-DA, LDA and RF. Most of the classification models showed acceptable results, while the identification rate was greater than 85%. The distinction between the healthy tobacco leaves and diseased ones resulted in classification accuracies of up to 95% with the BPNN and ELM models. [107,108]. Morellos et al. studied the early detection of tomato chlorosis virus (ToCV) in Belladonna cultivar tomato plants. The NCA algorithm was used for the effective wavelengths and most important SVI selection. The XY-F network and MLP–ARD ANN detected the ToCV infection and its severity level, scoring an overall accuracy of over 85%, with MLP–ARD performing generally better than XY-F [109].

Bienkowski et al. studied the possibility of the early detection of late blight and black leg and a variety of soilborn diseases (*R. solani*, *C. coccodes* and *S. subterranea*) on five different potato cultivars, Maris Piper, Estima, King Edward, Desiree and Mayan Gold, using either PLS and BPNN in greenhouse and field experiments. Unfortunately, the authors did not specify the phenotypic and genotypic differences of the cultivars or whether there was a difference in the important wavelengths for each cultivar. The models detected and distinguish diseases with obvious symptoms, even asymptomatic ones, correctly classifying the spectra from the greenhouse experiments with an accuracy of 84.6%. When the diseases were analyzed separately, the models were able to distinguish between the healthy and asymptomatic spectra leaves, plus three kinds of late blight with an accuracy of 92%. Models constructed with whole-plant reflectance data from the field had less accuracy [110].

Franceschini et al. studied early and late blight detection in three different tomato cultivars, Raja, Connect, and Carolus, with different degrees of resistance to late blight. Both UAV and ground-level data were used, including leaf analysis of chlorophyll content and canopy height. The important bands were chosen due to their importance in describing changes in the biochemical and biophysical traits of vegetation at the leaf and canopy levels. The relationship of leaf and vegetation pigment content was found to be less important than changes associated with structural traits. It was possible to identify considerable spectral changes related to late blight at early stages in between 2.5 and 5.0% of affected leaf area. The authors mentioned that the characteristics of different potato cultivars may potentially affect the spectral response, and they recommend considering it in future studies [111].

Gold et al. investigated the pre-symptomatic detection and differentiation of late blight and early blight in potato on four different potato cultivars: Katahdin, Snowden, SP951 and russet Burbank. The authors mention that cultivar features had a strong influence on spectral reflectance, but not on the visible reflectance range alone. The cultivars differed biochemical and physiological indices at different time stages of disease. The spectral responses of the four potato cultivars to infection were very different, yet they had important commonalities that made discrimination easier. Using the RF, PLS-DA, PCoA and NDSI approaches, the authors could distinguish the infected plants with greater than 80% accuracy two–four days before visible symptoms appeared. The individual stages of disease development for each pathogen could be distinguished from the corresponding control samples with accuracies of 89–95%. The authors reported that they could distinguish latent *Phytophthora infestans* from both latent and symptomatic *Alternaria solani* infection with greater than 75% accuracy. The spectral characteristics important for the detection of late blight changed during infection, while the spectral characteristics important for the detection of early blight remained unchanged, reflecting the different biological characteristics of the pathogens concerned. The authors mentioned that phenolics concentration may be important for detecting symptomatic late and early blight infections. Finally, the authors reported their belief that this study establishes that the host genotype has a significant influence on spectral reflectivity and, therefore, on the biochemical and physiological characteristics of plants exposed to infection by pathogens. [112,113]. A systematization of the reviewed materials is presented in Table 3.

### 2.4. Hyperspectral Remote Sensing of Wheat Diseases

In the world’s agriculture, wheat occupies a leading place; it is cultivated almost everywhere and is of great importance for the population of the entire globe. There are a number of harmful diseases, mainly of micromicetal origin, affecting this crop, such as scab caused by *Fusarium graminearum* and other *Fusarium* spp., yellow rust caused by *Puccinia striiformis*, brown rust caused by *Puccinia triticina*, powdery mildew caused by *Blumeria graminis f.* sp. *tritici* and others, which can cause serious losses to the yield [114,115,116]. Due to the importance of wheat there are many articles dedicated to the detection of its diseases with HRS. In this this review, we concentrate on articles describing the most researched diseases of one crop, namely wheat Fusarium head blight (FHB) and wheat yellow rust (YR). 

FHB or scab is a serious disease of cereal crops, such as wheat, rye, barley and oat, that may also affect other crops [117,118]. Affected grains rapidly lose mass and shrink, which results in high crop losses and quality reductions [119,120,121]. *Fusarium* genus fungi may produce dangerous mycotoxins that are harmful for humans and animals. These mycotoxins accumulate in living organisms and can enter the human diet along the food chain [122,123]. Due to the high economic and health harmfulness of this disease, it is very important to detect it at the early stages in order to apply timely and proper control measures. There are many species in the *Fusarium* genus, and their influence on host plants differs significantly in different environments, complicating the task of determining them.

The studies on determining FHB on wheat using HRS have been undertaken by a number of research teams from different countries. Delwiche et al. first studied the detection of FHB in three different wheat cultivars: Grandin, Gunner and oxen, reaching detection 83–98% accuracy. The models developed for just one variety were useless when applied to other varieties [124]. Barbedo et al. determined FHB (*F. graminearum* and *F. meridionale*) on wheat kernels with over 91% accuracy [125]. Mahlein et al. studied the *F. graminearum*, isolate S.19 and *F. culmorum* isolate 3.37 infestation on seven different wheat cultivars that had different resistances to the disease: Thasos, Triso, Passat, Scirocco, Chamsin, Taifun and Sonett (Descriptive List of Varieties, Bundessortenamt, Germany 2017). Using SVM, it was possible not only to differentiate between healthy and infected samples with accuracies of more than 76%, but to differentiate between *F. graminearum* and *F. culmorum*. The authors found it possible to use HRS for wheat FHB resistance phenotyping [126,127]. Ma et al. applied CWA in the detection of *F. graminearum* and obtained an overall accuracy of 88.7% [128]. Huang et al. obtained a detection accuracy of 75% with SVM optimized with a genetic algorithm, using correlation analysis and wavelet transform for the selection of important bands, vegetation indices and wavelet features [24]. Zhang et al. developed a new Fusarium disease index (FDI) after determining the best index from the existing indices with PLS regression, reaching 89.8 accuracy in detecting *F. graminearum* [129,130]. Whetton et al. studied FHB in the laboratory and field in the wheat cultivar Solstice, using PLSR to determine FHB from yellow rust in wheat and barley [131,132].

Baurigel et al. determined *F. culmorum* at early stages on wheat cultivar Taifun. The diseased and healthy wheat ear tissues spectra were differentiated with PCA. The authors noticed that spectral changes during disease development were based on variations in the content of carotenoids (500–533 nm) and, especially, that of chlorophylls (560–675 nm and 682–733 nm). Furthermore, spectral variations in the range of 927–931 nm reflected differences in the tissue water contents of healthy and diseased plant tissues. It was mentioned that the detection of FHB in the earliest stages is impossible due to missing symptoms. However, it was possible to detect FHB in later stages with the SAM method, with 91% accuracy. The mean detection accuracy was 67% during the whole study period (BBCH 65–89) [133].

Yellow rust (YR), in cereals, is a dangerous disease that can result in more than 60% yield shortage from outbreaks. The causative agent of YR is the mushroom *Puccinia striiformis* west, which affects more than 20 species of cultivated and wild cereals, including wheat, rye, triticale, barley and others. Until recently, the disease was of regional importance throughout the world. In 2000, the area of the pathogen expanded and its harmfulness increased [134,135,136,137]. Presently, YR epidemics can lead to extremely severe crop losses. It is very important to detect YR at the early stages in order to apply timely and proper control measures [138,139].

There are many studies dedicated to determining YR in wheat using HRS that have been undertaken by a number of research teams from different countries. Huang et al. used PRI to detect yellow rust in three different wheat cultivars, Jing 411, 98–100 and Xuezao, all differently resistant to yellow rust [140]. Zhang et al. studied another three cultivars, Jingdong8, Jing9428 and Zhongyou9507, and analyzed the relationship between nutrient stress and yellow rust injury, resulting in PhRI being the only index sensitive to yellow rust disease at all growth stages [141]. Krishna et al. used PLS, ANOVA and MLR to determine important bands and detect yellow rust with 92–96% accuracy. It was shown that the wheat crops affected by yellow rust have various symptoms and distinct spectra as compared with healthy ones [142]. Zhang et al. applied DCNN and RF to UAV data, wherein identification of the diseased and healthy plants was done by assessing NDVI data, reaching an overall accuracy of 85% [143]. Guo et al. used SVI (NRI, PHRI, GI and ARI) and spectral and textural features of hyperspectral images for YR detection in the wheat cultivar Mingxian 169, which is moderately susceptible to YR. The important bands were selected with SPA. During the research of wheat YR with UAV, the spatial resolution had lesser influence in the SVI methods, but significantly influenced the TF-based methods. The total YR detection accuracy in both studies was up to 95.8% [144,145]. Whetton et al. studied YR in the laboratory and field on the wheat cultivar Solstice using PLSR to determine FHB from yellow rust in wheat and barley [131,132].

The early detection of wheat YR was achieved by Bohnenkamp et al. who used two platforms with two different hyperspectral cameras: a ground-based vehicle (1–2-m height) and an UAV (20-m height). The data from the JB Asano cultivar, susceptible to YR, and the Bussard cultivar were analyzed with SAM and SVM to detect wheat YR. The most interesting moment in this study was the comparison of the hyperspectral data of the wheat canopy at the ground and UAV scales [146]. In another study Bohnenkamp et al. considered an interpretable decomposition of a spectral reflectance mixture under controlled laboratory conditions, studying a method of detecting and distinguishing between brown rust and yellow rust on the leaves of the wheat cultivars Taifun and Catargo. In this study, the authors took a very interesting approach to detecting spectral portraits of the pathogens themselves, studying the possibility of detecting YR uredinium on the surface of a wheat leaf, which made it possible to detect the disease at an early stage [147].

Zheng et al. were also able to solve the problem of early YR detection, continuing the research from [140] on the cultivars Jing 411, 98–100 and Xuezao, each differently resistant to yellow rust, by evaluating multiple different SVIs with LDA. Among those indices, SIPI, PRI, NDVI, PSRI, ARI, MSR, GI and NRI showed great potential for discriminating yellow rust disease in different growth stages, and PRI (570-, 525-, 705-nm ranges) for the early-mid stage and ARI (860-, 790-, 750-nm ranges) for the mid-late growth stage, were selected as the best spectral indices for monitoring yellow rust disease in wheat, with up to 93.2% classification accuracy [148]. A systematization of the reviewed materials is presented in Table 4.

### 2.5. Hyperspectral Remote Sensing of Other Crops and Their Diseases

A number of articles devoted to the early detection of diseases in various crops should also be mentioned, if without detailed analysis. These articles describe the detection of diseases such as: red leaf blotch on almond caused by *Polystigma amygdalinum* [149]; black sigatoka on bananas caused by *Mycosphaerella fijiensis* [150,151]; barley blast caused by *Magnaporthe oryzae* [152]; grapevine leafstripe [153]; grapevine leafroll caused by *Grapevine leafroll-associated virus 3* [154]; verticillium wilt of olive caused by *Verticillium dahliae* [155,156]; *Xylella fastidiosa* disease on olive trees [157]; peanut early leaf spot caused by *Cercospora arachidicola* S. Hori and late leaf spot caused by *Cercosporidium personatum* [158]; peanut bacterial wilt caused by *Ralstonia solanacearum* [159]; charcoal rot on soybean caused by *Macrophomina phaseolina* (Tassi) Goid [160] and corn leaf spot caused by *Phaeosphaeria maydis* (Henn.) [161].

### 2.6. Summary

The reviewed works prove the possibility of detecting oil palm [36,50,51,52,53,54,55,56,57,63,64,65,66], citrus [73,74,75,76,77,78], *Solanaceae* family crops [91,92,93,94,95,96,97,98,99,100,101,102,103] and wheat [24,124,125,126,127,128,129,130,131,132,140,141,142,143,144,145] diseases using HRS.

The possibility of detecting oil palm [58,59,60], citrus [79,80,81,82], *Solanaceae* family crops [104,105,106,107,108,109,110,111,112,113] and wheat [133,146,147,148] diseases at an asymptomatic, early stages was also proved.

Due to the peculiarities of oil palm and citrus crops, most of the experiments concerning these cultures were carried out in the field, except the research from Azmi et al. [60], which was conducted in a greenhouse under constant conditions, and of Li et al. [76] and Weng et al. [78], who conducted laboratory tests.

From Table 1 and the articles cited [56,57,60], it can be concluded that the most convenient range for detecting oil palm diseases is the interval from 600 to 950 nm, in which most of the authors’ were able to confidently determine the symptoms of oil palm diseases, including those in early stages. However, a strong scatter in the definition of important waves in various works allows us to conclude that there are unaccounted factors influencing their manifestation and thus selection. The specific BSR disease progression and its control methods also can influence the research direction. Therefore, it might be interesting to collect more data from oil palm seedlings in nurseries, laboratories and greenhouses, to avoid planting diseased seedlings.

From Table 2 it is obvious that the spread of the important waves for determining the diseases of citrus plants is even greater than in the case of the oil palm. The reason may be the physiological difference caused by the varietal diversity of citrus crops and various abiotic factors, which are mentioned in [76,78]. The study [78] describes the differences in the spectral portraits of different citrus cultures and the possibility of using the classification model developed for the Satsuma cultivar for HLB detection in the Ponkan cultivar by calibration model transfer. It would be interesting to apply similar technologies to other varieties of citrus fruits and other crops for which it is possible to identify diseases using HRS. It would also be interesting to conduct studies comparing the spectral portraits of different varieties and species of citrus fruits and detecting their abiotic stresses using a hyperspectral camera. A study of not only citrus tree canopies but also the citrus fruits themselves, as in the work of Qin et al. [162], is perhaps also an interesting and promising direction.

From Table 3, we can conclude that, at the moment, it is difficult to identify universal band ranges associated with certain diseases of *Solanaceae* family plants, even in the same culture. This may be related to the observations mentioned in [112,113], that the spectral responses from different cultivars are highly variable and that the host genotype has a significant impact on spectral reflectance, and, thus, on the biochemical and physiological traits of the plants undergoing pathogen infection. It is noteworthy that, in study [110], the authors did not mention the differences between spectral responses from the different cultivars that they describe. Meanwhile, judging by the data from [163], the described cultivars have phenotypical differences, which, theoretically, should entail a difference in the types of chlorosis caused by the studied diseases and, accordingly, in their spectral responses. These facts open up a large field for further studies of diseases in *Solanaceae* crops using HRS. It is probably necessary to pay more attention to the study of plants’ phenotyping and abiotic stress diversity using HRS, in order to subsequently facilitate the task of identifying those band ranges that are important for the early detection of *Solanaceae*-infecting diseases. We also believe that the search for patterns in spectral responses from different cultivars should start from those most similar in genotype and phenotype, thereafter studying more different ones. It will probably be possible to create data processing algorithms for the adaptation of ANNs trained on certain cultivars of *Solanaceae* crops to other varieties, different from them, as mentioned in [78]. It would be expedient to focus efforts on laboratory research and industrial greenhouses, since this will eliminate or significantly reduce the influence of abiotic factors on experiments, which, in the future, will create a basis for obtaining stable, repeatable results in field experiments.

From Table 4 we can see that, at the moment, as in the case of *Solanaceae* it is difficult to identify universal band ranges associated with certain diseases of wheat. It should be noted that in articles dedicated to wheat disease detection with HRS from all other crops, the authors pay the greatest attention not only to the variety of cultivars of the studied culture, but also to the factors of plant resistance and pathogen diversity and take into account their possible influences on the results of their experiments [124,125,126,127,131,132,133,140,141,145,146,147,148]. We also want to note an interesting approach to detecting the spectral portraits of YR pathogens themselves, described in [147]. In general, the analysis of articles devoted to wheat diseases—As the most studied crop—Allowed us to finally identify the main gaps in the field of early detection of plant diseases with HRS.

## 3. Discussion

We believe that, due to the lack of interaction between specialists in engineering and biology, there is a significant gap in the scientific basis for planning an experiment to use remote sensing data in determining plant state. Although the review above demonstrates the practical possibility of late and early detection of plant diseases using HRS, it also reveals differences in the technical results (range of important bands) between researchers, which indicates an insufficient study of the experimental methodology, as can be seen from Table 1, Table 2, Table 3 and Table 4.

As a result of hyperspectral remote sensing, for each pixel of a scene, we get a random vector, which can be considered the result of a random experiment. The outcome of a random experiment can be favorable or unfavorable, which is associated with the detection or non-detection of a disease in the space reflected by a particular pixel. Accordingly, these vectors can be processed by methods developed in the theory of probability and in mathematical statistics, which make it possible to effectively determine the characteristics of a random experiment. In this case, the tasks of data normalization and the allocation of those frequency bands (important bands) that make the greatest contribution to the outcomes of experiments (favorable or unfavorable) and, accordingly, are the most informative for identifying diseases, can be solved. The selection of important bands is a critical step in the detection of plant diseases using HRS. As a rule, data normalization is carried out first to get rid of noise. Then, various algorithms are applied to identify important bands, such as Savitzky–Golay filtering [50,51,58,81,82,83,99,100,109,131,132,147]; the Mann–Whitney U test [52,54]; coefficient of variation [60]; PCA [74,76,79,92,95,99,109,133]; SPA [78,102,106,107,108,144]; GA and BRT [106]; SAM [112,113,129,146].

The listed algorithms make it possible to achieve the determination of important bands. Various methods of machine learning allow achieving a fairly high accuracy in identifying diseases (between 60 and 95% accuracy) based on those data. However, from Table 1, Table 2, Table 3 and Table 4, we can conclude that even under very similar experimental conditions—For example when studying oil palms—Different sets of important bands are obtained at the output, often with a spread of more than 100 nm [52,53,54,55,56,57,58,59,60]. Xie et al., in [103], used five different algorithms to select important bands, taken from five different studies: *t*-test [164], Kullback–Leibler divergence [165], Chernoff bound [166], receiver operating characteristics [167] and the Wilcoxon test [168]. It is noteworthy that, in 4 tests out of 5, only 1 frequency out of 15 matched closely. In this case, the scatter of the ranges of all initially selected important bands was in the range from 400 to 850 nm, (400, 402, 403, 411, 413, 418, 419, 420, 422, 473, 642, 690, 722, 756 and 850 nm), i.e., practically in the entire range of the used sensor (380–1020 nm).

Based on the data from Table 1, Table 2, Table 3 and Table 4, we assume that, in the experiments on the same section of a field, repeated in different years or seasons, different important bands will likely be allocated when using automatic selection methods. Unfortunately, at the moment it is not possible to test this theory, since there are very few articles in which such experiments would be described.

Summarizing the topic of choosing the important bands for plant disease detection, we assume that it would be logical to focus on studying the bands of biochemical changes occurring in diseased plants and screening out the bands not related to the given disease, rather than using machine learning.

To successfully conduct the biological component of experiments on the HRS of plant diseases, it is necessary to understand that plant diseases are a particular case of plant stress. Plant diseases are processes that occur in plants under the influences of various reasons and which lead to their oppression and decreased productivity. Plant diseases are divided into two main groups: infectious and non-infectious [29,30]. The infectious plant diseases are caused by microorganisms (mainly fungi, bacteria, viruses and nematodes) or parasitic plants. The non-infectious diseases can be caused by genetic disorders or physiological metabolic disorders resulting from unfavorable environmental conditions [29,30]. Plant diseases almost always have visible symptoms that we can observe in a certain spectral range. In their early stages, such symptoms appear in the form of various chloroses or, less often, necrosis or pustules, with a huge variety of manifestations [169,170]. In the case of an asymptomatic course of the disease in its early stages, for example barley Ramularia disease caused by *Ramularia collo-cygni* [171], Fusarium head blight of different cereals caused by *Fusarium culmorum* [133] or soybean Sudden death caused by *Fusarium virguliforme* [172], early detection by remote sensing can be challenging.

Plant stress is a state of the plant in which it is influenced by unfavorable abiotic (light, heat, air, humidity, soil composition and relief conditions) and biotic factors (phytogenic, zoogenic, microbogenic and mycogenic). Plant responses to both abiotic and biotic stress is usually complex and includes both nonspecific (common for different stressors) and specific components. In a state of stress plants stop their growth, sharply reduce the activity of their root systems and reduce the intensity of photosynthesis and protein synthesis [173,174,175]. In a significant number of stressful situations, an immune response causes an increase of certain metabolites content, such as jasmonates or salicylates [175,176,177,178,179,180]. These reactions can be detected using hyperspectral sensors [181,182,183,184,185,186,187,188]. The study of plant stress using hyperspectral sensors is presented in a number of works [189,190,191], including those comparing the spectral portraits of plants simultaneously exposed to biotic and abiotic stress [192,193,194,195]. It is necessary to take into account many abiotic factors in addition to the possible influence of pathogens to accurately determine the reasons for stress manifestation [59,60,63,73,78,92,98,112,113,124,126,127,133,141]. Our analysis indicates that there is no unified methodology for conducting hyperspectral studies of plant diseases that takes into account the influence of abiotic factors. That is why we believe it is best to carry out experiments in laboratory conditions or in industrial greenhouses in order to partially or completely eliminate abiotic factors. Attempts to create various mobile vehicles operating at ground level whose purpose is to replace natural light sources with artificial light when using hyperspectral sensors in field experiments are described in [73,74,75,92,94,123]. This solves one of the main problems associated with the inhomogeneity of the solar spectrum due to changing weather conditions. Nevertheless, this approach cannot completely solve the problem of the influence of abiotic factors.

It would also be interesting to continue studies describing the definition of the phenotype and/or genotype of a plant and its influence on changes in the spectral portrait thereof [196,197,198,199,200,201]. Several studies reviewed describe that the host plant genotype has a significant impact on spectral reflectance and on the biochemical and physiological traits of the plants undergoing pathogen infection [76,78,110,111,112,113,124,126,127,140,141,147,148]. Therefore, it is very important to indicate the culture and cultivar of the studied plants. The exact indication of pathogens used for inoculation is also very important. We believe that comparisons of the spectral portraits of plants of different cultivars of the same crop is a primary task in creating a general methodology for detecting plant diseases using hyperspectral sensors. It is possible that the influence of chlorophyll fluorescence on the spectral portraits of plants and their related SVI may be a significant contribution to the solution of this problem [155,202,203,204,205]. Success in this area may allow the creation of patterns for determining phenotypes and plant cultivars within one crop, which will become the basis for a database of hyperspectral portraits of plants.

If we can confidently detect different types of plant stresses and distinguish plants infected with pathogens from healthy one and/or those affected by abiotic stresses, we can study the influence of the genotypic characteristics of a pathogen on the spectral profile of an infected plant. To do this, it is necessary to identify the differences between plants of the same phenotype as affected by pathogens with different genotypes. Since, for many pathogens, primarily micromycetes, the intrageneric and even intraspecific diversity is extremely high, it is necessary to investigate the possible differences in the spectral manifestations of symptoms, for example, between different species of fungi of the genus *Fusarium* or between different races of the brown rust pathogen (*Puccinia triticina*). The aim of such experiment will be to study the effect of the phenotypic and genotypic diversity of pathogens on the variability of spectral portraits of host plants. The visual manifestations of symptoms of yellow rust (*Puccinia striiformis*) caused by different races or different strains of *Fusarium graminearum* are often very similar. In the early stages of the disease, chlorosis caused by pathogens of different species may have similar spectral portraits, which become more distinguishable in the later stages of the disease, and, thus, is also an important direction for research [91,92,96,97,102,103,110,111,112,125,126,127,131,132]. The influence of plant resistance on the symptomatology of pathogenesis and works describing the difference in the data obtained in such cases is also worth mentioning [110,111,112,113,126,127,132,140,141,144,145,146,148]. The determination of resistant cultivars using hyperspectral sensing is also a promising area of research with great applied potential [126].

One more direction, which is important for the early detection of plant diseases using HRS, is the study of spectral portraits of pathogens themselves. Unfortunately, this is only possible for a small number of diseases, such as wheat powdery mildew caused by *Blumeria graminis* and wheat yellow rust of wheat caused by *Puccinia striiformis*, which show characteristic external symptoms in the early stages. Usually, these are diseases of fungal origin, where the object of detection is micromycete mycelium or spores on the leaf surface of a diseased plant. Disease detection by this method is considered in the example of wheat yellow rust, using pure fungal spore spectra as reference [147].

Pest control is also an important aspect of plant protection. We hypothesize that HRS can also be used to early detect such dangerous pests as the Colorado potato beetle (*Leptinotarsa decemlineata*), sunn pest (*Eurygaster integriceps*) [206], or western corn rootworm (*Diabrotica virgifera virgifera*), using spectral portraits of imago and different ages of larvae. Currently, a small number of works have been published on this topic [191,206,207,208,209,210], but we consider this direction to be very promising, especially for use in industrial greenhouses. Another possible direction of research is the detection of local outbreaks of pests outside farmlands, for example, locusts (*Acridoidea*) or beet webworms (*Loxostege sticticalis*), in order to eliminate them early before these pests can cause damage to yields.

We believe that the effect of biochemical changes in plant tissues is critical for the early detection of plant diseases using passive sensors. The reflectance of light from plants leaves is dependent on multiple biophysical and biochemical interactions. The VIS range (400–700 nm) is influenced by pigment content. The NIR range (700–1100 nm) is influenced by leaf structure, internal scattering processes and by the light absorption by leaf water. The SWIR range (1100–2500) is influenced by chemicals and water composition [196,211,212,213,214,215,216].

The most investigated areas in this topic are the determination of changes in the content of water, nitrogen (N) in plants, as well as of chlorophyll or carotenoids, using various SVIs, which can be used to detect plant diseases. These techniques can be used to determine the nitrogen content of plants [217,218,219] and to detect plant stresses and diseases [56,57,78,220,221,222], including the early detection of plant diseases and pest infestations [147,154,156,157,223].

The topic of detecting individual chemical elements or chemical compounds, including volatiles, in plants is a less studied problem. In plant physiology, such elements are of great importance, such as nitrogen (N), one of the key components for chlorophyll; phosphorus (in the monovalent orthophosphate form H_2_PO_4_^−^), a key macronutrient; potassium (K^+^), influencing leaf color; calcium (Ca^2+^), which plays a fundamental physiological role in leaf structure and signaling; magnesium (Mg^2+^), an essential macronutrient for photosynthesis (as it is the central atom of chlorophyll); sulfur (S), in the form of sulfate; iron (Fe^2+^ or Fe^3+^), copper (Cu^2+^), manganese (Mn^2+^) and zinc (Zn^2+^), which are essential elements for plant growth and components of many enzymes; and the ions responsible for salination: Na^+^, K^+^, Ca^2+^, Mg^2+^ and Cl^−^ [216]. The detection of these elements by HRS can be a key factor for identifying plant diseases at an early stage, since plant diseases are accompanied by a deficiency of some of the listed elements, which is the cause of chlorotic and necrotic changes in plant tissues [216]. Unfortunately, this task is difficult and poorly studied, but the following works prove the possibility of determining the chemical composition of plants in the VIS, NIR and SWIR ranges. Pandey et al. detected a wide range of macronutrients, namely N, P, K, Mg, Ca and S, and micronutrients, namely Fe, Mn, Cu and Zn, in maize and soybean plants [224]. Zhou et al. detected cadmium (Cd) concentrations in brown rice before harvest [225]. Ge et al. tried to analyze chlorophyll content (CHL), leaf water content (LWC), specific leaf area (SLA), nitrogen (N), phosphorus (P) and potassium (K) in maize using different SVIs but succeeded only with CHL and N [226]. Hu et al. proved to determine the content of Ca, Mg, Mo and Zn in wheat kernels [227].

The most difficult and interesting direction is the detection of the content of not individual elements, but more complex chemical compounds using HRS. As an example of such works, one can cite the articles by Gold et al., where the mechanisms of physiological changes in potato plants were considered when inoculated by *Alternaria solani* and *Phytophthora infestans* pathogens in the analytical example of the contents of foliar nitrogen, total phenolics, sugar and starch [112,113]. Fuentes et al. monitored the chemical fingerprints of different leaf samples and studied the correlation of aphid numbers in wheat plants with the presence and quantity alcohol, methane, hydrogen peroxide, aromatic compounds and amide functional groups compounds [228]. The paper [228] presented results on the implementation of SWIR HRS (1596–2396 nm) and a low-cost electronic nose (e-nose) coupled with machine learning. The authors believe that such study of plant physiology models open their use to assessing models of other biotic and abiotic stress effects on plants. Thus, the search for plant diseases at early stages using passive sensors, including hyperspectral ones, should be carried out in three main directions: the search for the characteristic immune response of the host plant to the pathogen, the search for characteristic symptoms of plant damage by the pathogen or the search for spectral portraits of the pathogen or pest itself. It is always necessary to take into account other stress factors affecting the spectral portrait of a diseased plant, which will allow us to accurately determine plant diseases using passive remote sensing.

Further development of experiment planning should be considered, preferably using a common methodology, so that there is an opportunity to adequately compare the results. An experiment tree, which will consider the physiological parameters of the plant should be designed [229]. All phases of the experiment should be considered and planned in advance, on the basis of the science of experiment planning, which is sufficiently well developed for applied physical research, based on the methods of probability theory and mathematical statistics. The following research phases for each type of sensors should be developed: laboratory research in deterministic conditions of deterministic parameters; the allocation of spectral bands responsible for certain parameters of plants (including diseases) in laboratory experiments; repetition (possibly multiple) of a laboratory experiment to collect statistics and validate; transfer of the experiment to field conditions to verify the correctness of the selected spectral bands. Such planning of experiments and the creation of a methodology for conducting them fills in the gaps associated with the lack of consideration of such factors as: different phenotypes of plants and their different spectral responses; various diseases and also their different spectral responses; the need to create and take into account a model of light propagation from an irradiating source to normalize hyperspectral imagery data [229,230,231,232].

It would be interesting to see more data comparing datasets collected from the same crops with different models of hyperspectral sensors. There are several articles that mention the use of two different sensors during the same experiment [76,80,147], but there is no data on how sensor model can affect data variability. The different types of hyperspectral sensors, i.e., spectroradiometers and hyperspectral cameras, have their own strengths and weaknesses, and experiments are needed to compare the results obtained from their usage. It is mentioned that a spectrometer device has a limitation when compared with a camera, where it can only take one reading per time for a small sample point, thus requiring a longer duration of data collection [60], but this should not affect the outcomes of experiments. It is assumed that the spectral portraits of plants should be the same regardless of the sensor model and type, which should allow developing a unified platform for the early detection of plant diseases. We believe that it is also possible, together with the use of hyperspectral sensors, to use active sensors in laboratory studies, which are successfully used to determine plant diseases, such as Raman spectrometers [233,234]. Comparison of spectral portraits obtained from the same samples using two different types of sensors may help to understand which factors most strongly affect hyperspectral portraits and to either make appropriate changes to the experiments or to create algorithms for correcting hyperspectral portraits. Such approach was used by Mahlein et al. in study [127], wherein HRS data are compared with those of chlorophyll fluorescence and thermal sensors, and by Fuentes et al. in study [228], wherein an electronic nose was used to determine the content of certain volatile chemical compounds to refine the HRS data in a SWIR diapason.

We have summarized the available data in a pivot table (Table 5), from which the following conclusions can be made. Spectrometers used without a connection to a photo camera have the least efficiency, both in obtaining a high percentage of detection of diseased plants and in determining the early stages of diseases. However, when used in conjunction with a photo camera, their effectiveness increases significantly. Hyperspectral cameras have the highest percentage of use for early detection and good results in obtaining a high percentage of detection of diseased plants. We can conclude that these results indicate that it is better to use hyperspectral cameras or a combination of a photo camera and a spectrograph to study plant diseases in the early stages. In the future, it is possible to switch to using a combination of a photo camera and a spectrograph for practical purposes, since this solution is economically more profitable.

The topic of HRS under consideration is quite new, so we did not add to this comparative analysis table (Table 5) data on the number of articles in which, from our point of view, the technical and physical parts of an experiment are correctly stated, which is the subject of a separate discussion.

We hope that the analysis carried out in this review of the main errors and gaps will help solve problems regarding experiment planning and undertaking.

The main disadvantage (which should be mentioned separately, since almost all the articles under consideration contain it) is the lack of repeatability in the experiments performed. It is critical for scientific validity to run an experiment at least twice. If we are talking about a field experiment, then a repeated experiment is carried out, as a rule, in the next growing season. In the laboratory, the experiment is carried out at least twice, and the test of the effectiveness of training any AI algorithms used by researchers should be carried out on a second dataset without additional training. Only if such experiment is successful can we talk about the scientific nature of the results thereof and its success in detecting plant disease. We also want to repeat the importance of understanding the physiology of the processes occurring in a diseased plant, since, from our point of view, the chemical composition of the tissues of diseased plants is of primary importance for the selection of the ranges of important bands for determining disease. These ranges should be very similar for phenotypically similar plants of the same species, however, from Table 1, Table 2, Table 3 and Table 4 it can be seen that there are practically no exact matches of important bands. In any case, we believe that such coincidences are insufficient.

Additionally, as we have mentioned earlier, the methods of analyzing the data obtained (machine learning, neural networks, statistical analysis, manual analysis), in our opinion, are only methods of automation that do not make a significant contribution to solving the problem of the early detection of plant diseases with HRS [90,91,92,93,94,97,105,142,143].

The definition of plant diseases with remote sensing cannot be considered in isolation from other parameters and related factors—i.e., the phase of plant development, phenotype, multiple external factors. Therefore, the main task that needs to be addressed when using hyperspectral imaging for early detection of plant diseases, in our opinion, is the application of a systematic approach. That is, determining the place in a complex natural–technical system at which it is necessary to analyze the elements of the system and their interrelationships within the framework of a specific organizational structure to detect violations of this structure (that is, plant parameters violations during development).

Summing up our review, we would like to point out the articles that, in our opinion, best describe certain aspects of this problem in relation to various plant crops [35,60,76,78,79,80,104,105,109,110,113,126,127,133,146,147,192,203,213]. We would especially like to acknowledge the work of a team of authors from the Institute of Crop Science and Resource Conservation (INRES) Plant Diseases and Plant Protection, University of Bonn [2,35,126,127,146,147,182,196,197]. We believe that these works are the most relevant, the most widely disclosing of the topic and which offer the greatest number of interesting solutions and new approaches.

## 4. Conclusions

At the moment, a sufficient number of articles are available in the field of using HRS to conduct successful experiments for the early detection of plant diseases. From the articles reviewed, it is also clear that the usage of machine learning methods for early detection requires a significant HRS source data of plant diseases.

We have made a number of assumptions about possible knowledge gaps preventing the successful replication of experiments in plant disease detection using HRS. Differences in spectral portraits can be caused by various abiotic and biotic factors that cause plant stress. The manifestation of disease can also be influenced by the phenotype or genotype of the host plant, which determine the level of plant resistance. The presence of a mixed infection may also be an important factor influencing the spectral portrait. Finally, last but not least, it is important to understand the biochemical changes that occur within the plant during stress, in what wave range they manifest themselves and how they may affect the spectral portrait of the plant.

In terms of technical and physical aspects, it is necessary to consider the propagation model of sunlight when conducting field experiments or to consider the characteristics of artificial light sources used when conducting laboratory experiments, because HRS is a passive remote sensing method that depends on external light-source conditions. Other technical issues may include the incorrect use of the equipment. The correct calibration of a hyperspectral sensor or camera is needed for proper data collection. Such calibration depends on hyperspectral sensor temperature; thus, the equipment must be recalibrated after some continuous period of work due to its warming during operation. It is best if the experiment description includes information about the times of day when the HRS data was obtained.

We have reached the main objective of our study by proving the possibility of early plant disease detection by hyperspectral remote sensing. Our assumption about the coincidence of important bands for the same diseases and plants is partially proved by the results of the reviewed articles. From the other side, many of the reviewed articles demonstrate a mismatch of such bands, which highlights one of the found methodological gaps—That a model of light propagation in different conditions should be developed to normalize data obtained thereawith.

The systematization of modern relevant works about the early detection of plant diseases is present in the Discussion and in Table 1, Table 2, Table 3, Table 4 and Table 5.

In our opinion, the direction of further research should be as follows. To successfully solve the problem of the early detection of plant diseases using HRS, the joint work of specialists in plant physiology, phytopathology, plant resistance, phenomics, bioinformatics, information technologies, system analysis and optics or photonics is required. We also believe that, at this stage of research development, it is more logical to conduct experiments in laboratory conditions or in industrial greenhouses of the latest generation, where the variability of abiotic factors is minimized. The creation of databases of hyperspectral portraits of plants of various crops and cultivars exposed to the influence of various pathogens should probably be postponed until the general principles of remote sensing of plant diseases using HRS are developed to avoid possible confusion.

## Figures and Tables

**Figure 1 sensors-22-00757-f001:**
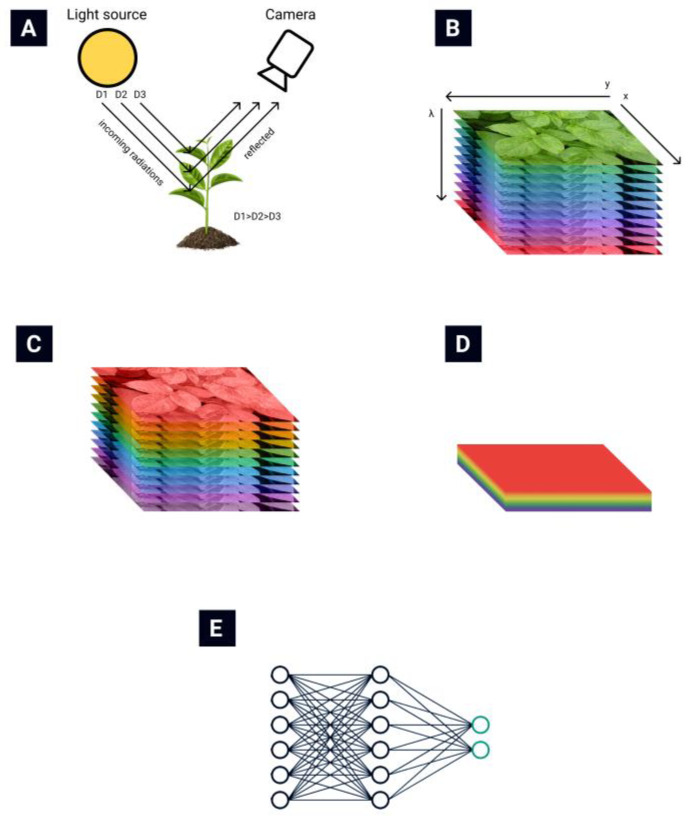
Hyperspectral data retrieval and processing (remastered from [14,15]). (**A**) Reflected light collection by the hyperspectral camera, (**B**) a hyperspectral data cube, (**C**) data normalization, (**D**) feature extraction, (**E**) automation of the classification process.

**Table 1 sensors-22-00757-t001:** Oil palm disease early detection by HRS.

Publication Year	Culture	Treat	Equipment	Studied Bands	Important Bands	Study Type	Reference	Location
2009	oil palm	basal stem rot	APOGEE spectroradiometer of unmentioned model	450–1100	715, 734, 791	field	[52]	Malaysia
2009	oil palm	basal stem rot	APOGEE spectroradiometer of unmentioned model	300–1000	462, 487, 610.5, 738, 749	field	[53]	Malaysia
2010	oil palm	basal stem rot	PP Systems Unispec-SC spectrometer	310–1130	670–715, 490–520, 730–770, 920–970	field	[50,51]	Indonesia
2011	oil palm	basal stem rot	APOGEE spectroradiometer of unmentioned model	350–1000	495, 495.5, 496, 651.5, 652, 652.5, 653, 653.5, 654, 654.5, 655, 655.5, 656, 656.5, 657, 657.5, 658, 658.5, 659, 659.5, 660, 660.5, 661, 908	field	[55]	Malaysia
2014	oil palm	basal stem rot	ASD spectrometer of unmentioned model	325–1040	not mentioned	field	[58]	Malaysia
2017	oil palm	basal stem rot	APOGEE spectroradiometer of unmentioned model	325–1000	495, 495.5, 496, 651.5, 652, 652.5, 653, 653.5, 654, 654.5, 655, 655.5, 656, 656.5, 657, 657.5, 658, 658.5, 659, 659.5, 660, 660.5, 661, 908	field	[56]	Malaysia
2017	oil palm	basal stem rot	GER 1500 spectrometer	273–1100	540–560, 650–780	field	[59]	Malaysia
2018	oil palm	basal stem rot	Specim spectrograph of unmentioned model	350–1000	650–750	field	[57]	Malaysia
2020	oil palm	basal stem rot	Cubert S185 camera	325–1075	800–950	greenhouse	[60]	Malaysia
2014	oil palm	orange spotting	ASD FieldSpec 4 spectrometer	300–1050	400–401, 404–405, 455–499, 500–599, 600–699, 700–712	field	[63,64]	Malaysia
2019	oil palm	orange spotting	ASD HandHeld 2 spectrometer	400–1050	601–630	field	[36]	Malaysia
2019	oil palm	orange spotting	ASD HandHeld 2 spectrometer	325–1075	680–780	field	[65,66]	Malaysia

**Table 2 sensors-22-00757-t002:** Citrus disease early detection by HRS.

Publication Year	Culture	Treat	Equipment	Studied Bands	Important Bands	Study Type	Reference	Location
2012	citrus	citrus greening	Spectra Vista SVC HR-1024 spectrometer	350–2500	537, 612, 638, 662, 688, 713, 763, 813, 998, 1066, 1120, 1148, 1296, 1445, 1472, 1546, 1597, 1622, 1746, 1898, 2121, 2172, 2348, 2471, 2493	field	[73,74,75]	USA
2012	citrus (orange)	citrus greening	Spectra Vista SVC HR-1024 spectrometer & Varian Cary 500 Scan	457–921	650–850	field and lab	[76]	USA
2012	citrus (orange)	citrus greening	Specim Aisa Eagle camera	457–921	410–432, 440–509, 634–686, 734–927, 932, 951, 975, 980	field	[77]	USA
2018	citrus	citrus greening	Specim Imspector V10E spectrograph combined with camera	379–1023	493, 515, 665, 716, 739	lab	[78]	China
2019	citrus	citrus greening	Cubert S185 camera and ASD HandHeld 2 spectrometer	400–1000	544, 718, 753, 760, 764, 930, 938, 943, 951, 969, 985, 998, 999	field	[80]	China
2020	citrus	citrus greening	Cubert S185 camera & ASD HandHeld 2 spectrometer	450–950, 325–1075	468, 504, 512, 516, 528, 536, 632, 680, 688, 852	field	[79]	China
2020	citrus	citrus greening	ASD HandHeld 2 spectrometer	370–1000	not mentioned	field	[83]	China

**Table 3 sensors-22-00757-t003:** *Solanaceae* disease early detection by HRS.

Publication Year	Culture	Treat	Equipment	Studied Bands	Important Bands	Study Type	Reference	Location
2003	tomato	late blight	Megatech GER-2600 spectrometer	400–2500	750–930, 950–1030, 1040–1130	field	[98]	USA
2014	tobacco	TSWV	Ocean Optics USB2000 spectrometer	450–850	475.22, 489.37, 524.29, 539.65, 552.82, 667.33, 703.56, 719.31, 724.31, 758.39	greenhouse	[101]	Bulgaria
2015	tomato	late blight, early blight	Specim Imspector V10E spectrograph combined with camera	400–1000	442, 508, 573, 696, 715	lab	[102]	China
2017	tomato	gray mold	Specim Imspector V10E spectrograph combined with camera	380–1023	655, 746, 759–761	lab	[103]	China
2017	tomato	yellow leaf curl	Specim Imspector V10E spectrograph combined with camera	450–1000	560–575, 712–729, 750–950	lab	[91]	China
2017	tobacco	TMV	Specim Imspector V10E spectrograph combined with camera	450–1000	697.44, 639.04, 938.22, 719.15, 749.90, 874.91, 459.58, 971.78	lab, greenhouse	[107,108]	China
2018	tomato	late blight, target and bacterial spot	Spectra Vista SVC HR-1024 spectrometer	350–2500	445, 450, 690, 707, 750, 800, 1070, 1200	lab	[92]	USA
2018	tomato	TSWV	Specim Imspector V10E spectrograph combined with camera	400–1000	700–1000	lab	[104]	Israel
2018	potato	PVY	ASD FieldSpec 4 spectrometer	350–2500	500–900, 720–1300	field	[94]	USA
2019	tomato	late blight, blackleg	StellarNet Blue Wave spectrometer	400–1000	not mentioned	greenhouse, field	[110]	UK
2019	tobacco	TSWV	Surface optics SOC710VP camera	400–1000	780–1000	lab	[106]	China
2019	potato	PVY	Specim FX10 camera	400–1000	not mentioned	field	[93]	The Netherlands
2019	potato	early blight	Specim Imspector V10E spectrograph combined with camera	430–900	550, 680, 720–750	field	[95]	Belgium
2019	tomato	bacterial spot, target spot	Resonon Pika L camera	380–1020	408–420, 630–650, 730–750	lab and field	[97]	USA
2019	pepper early	TSWV	Specim Imspector V10E spectrograph combined with a camera	400–1000	700–1000	lab	[105]	Israel
2019	potato	late blight	Senop Oy Rikola camera	500–900	620, 724, 803	field	[111]	The Netherlands
2020	tomato	yellow leaf curl, bacterial spot	Resonon Pika L camera	380–1020	550–850	lab and field	[97]	USA
2020	tomato early	ToCV	PP Systems Unispec-SC spectrometer	310–1100	402.2, 405.5, 412.2, 415.6, 425.7, 429.0, 449.2, 556.4, 559.7, 563.0, 566.4, 676.4, 679.7, 722.9, 726.3, 862.1	lab	[109]	Greece
2020	potato	late blight	ASD FieldSpec 4 spectrometer	400–900	439–481, 554–559, 654–671, 702–709	lab	[99]	Canada
2020	potato	late blight	ASD FieldSpec 4 spectrometer	660–780	668, 705, 717, 740	lab	[100]	Canada
2020	potato early	late blight, early blight	Spectra Vista SVC HR-1024 spectrometer	350–2500	700, 857, 970, 990, 1100, 1241, 1380, 1890, 2300	lab	[112,113]	USA

**Table 4 sensors-22-00757-t004:** Wheat disease early detection by HRS.

Publication Year	Culture	Treat	Equipment	Studied Bands	Important Bands	Study Type	Reference	Location
2000	wheat	fusarium	Specim Imspector V9 spectrometer combined with camera	425–860	not mentioned	lab	[124]	USA
2011	wheat	fusarium	Specim Imspector V10E spectrograph combined with camera	400–1000	500–533, 560–675, 682–733	lab and field	[133]	Germany
2015	wheat	fusarium	Headwall Photonics Hyperspec Model 1003B-10151 spectrometer combined with a camera	520–1785	1411	lab	[125]	Brazil
2018	wheat	fusarium	Specim Imspector V10E and ImSpector N25E spectrographs	400–1000, 1000–2500	430–525, 560–710, 1115–2500	greenhouse	[126,127]	Germany
2018	wheat	fusarium, yellow rust	Gilden Photonics camera	400–1000	650–700	lab, field	[131,132]	UK
2019	wheat	fusarium	ASD FieldSpec Pro spectrometer	350–2500	471, 696, 841, 963, 1069, 2272	field	[128]	China
2019	wheat	fusarium	Surface optics SOC710VP camera	400–1000	447, 539, 668, 673	field	[129]	China
2020	wheat	fusarium	Surface optics SOC710VP camera	400–1000	560, 565, 570, 661, 663, 678	field	[130]	China
2020	wheat	fusarium	ASD FieldSpec Pro spectrometer	350–2500	350–400, 500–600, 720–1000	field	[24]	China
2007	wheat	yellow rust	ASD FieldSpec Pro spectrometer	350–2500	not mentioned	field	[140]	China
2012	wheat	yellow rust	ASD FieldSpec Pro spectrometer	350–2500	not mentioned	field	[141]	China
2014	wheat	yellow rust	ASD FieldSpec Pro spectrometer	350–2500	428, 672, 1399	field	[142]	India
2019	wheat	yellow rust	ASD FieldSpec Pro spectrometer	350–1000	460–720, 568–709, 725–1000	field	[148]	China
2019	wheat	yellow rust	Specim ImSpector PFD V10E camera, Senop Oy Rikola camera	400–1000, 500–900	594, 601, 706, 780, 797, 874, 881	field	[146,147]	Germany
2019	wheat	yellow rust	Cubert S185 camera	450–950	not mentioned	field	[143]	China
2019	wheat	yellow rust	Headwall Photonics VNIR imaging sensor, Cubert S185 camera	400–1000	538, 598, 689, 702, 751, 895	lab, field	[144,145]	China

**Table 5 sensors-22-00757-t005:** Comparative analysis of hyperspectral remote sensing usage.

	Spectrometer	Spectrometer with a Camera	Hyperspectral Camera
total articles	32	13	16
early detection	8	4	6
total/early ratio	25%	33.33%	37.50%
accuracy > 90%	15	13	11
accuracy > 90%/total articles ratio	46.88%	100%	68.75%

## Data Availability

Not applicable.

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
