# Peer review of "Current State of Hyperspectral Remote Sensing for Early Plant Disease Detection: A Review"

_sensors, 2022, doi:10.3390/s22030757_

Round 1

Reviewer 1 Report

Contribution: The manuscript is a literature review of hyperspectral remote sensing techniques for early plant disease detection. It includes nine pages of tables and eight pages of references. However, only diseases of four culture types are analyzed: oil palm, citrus, vegetables, and wheat.

Organization and comprehension: Hyperspectral remote sensing methodology and techniques are not described at all. Please add a section describing basics of the hyperspectral remote sensing and include some illustrations. A list of abbreviations appears at the end of the manuscript. In spite of that, please define the abbreviation in the text (for example, page 1, integrated pest management (IPM))...

Scientific approach: This kind of manuscript (a review) must include analysis and comparison (in one word, criticism) of the reviewed HRS techniques. Please provide a few tables in the "Discussion" section presenting advantages and disadvantages of the described techniques and equipment in Table 1, Table 2, Table 3, and Table 4. 

Language: Please correct grammar mistakes. For example, "systematic table of ... is present(ed)" - Abstract; "The possibility of detecting ... has also being (been) proved" - page 4...

Author Response

Dear reviewer, the authors are grateful for your comments, which helped us improve our manuscript.

Please find the answers to your questions in the attached file.

Reviewer 2 Report

The rapid spread of various plant diseases is one of the important problems facing modern agriculture. Remote sensing, especially the hyperspectral image with detailed spectrum of crops, shows great potential for diseases early monitor. So, it is very important to review the development of hyperspectral remote sensing in the crop application.

I have to say, the authors collect many references in this field including different crops, such as oil palm, citruses, Solanaceae family plants and wheat, and different types of diseases using different types of hyperspectral sensors. However, in my opinion, the authors did little job about the review process, so it seems like a report not a review. Here I suggest the authors carefully analyze the references, and improve the review paper.  The readers would have more interests if the disease of mostly cultivated crop, wheat, are carefully studied from the sensors, method, products to systems, etc.

Specific comments:

  • In the Introduction section, I could not conclude the logic between paragraphs, and some opinions are lack of enough evidence, such as “We believe that, despite the advances in CNN architecture, this approach for …there are limitations associated with the small size of training sets available”.
  • In the Materials and Methods, the same pattern is applied for four different crops, same subtitle, same structure, even same table.
  • I strongly suggest the authors focus on one crop, such as wheat, and make a clear and deep review about hyperspectral application from sensors to method, from past to future, from basic to state-of-the-art.

Author Response

(The authors gave the same response as above.)

Round 2

Reviewer 1 Report

My concerns regarding Remark 1 (Contribution) and Remark 4 (Scientific approach) still stand. There is no significant improvement of the manuscript.

Author Response

Dear reviewer!

Please find the answers to your questions in attached file.

Reviewer 2 Report

The manuscript has been improved after the modification. Although I don't like the whole structure, I do not mind it is published.

Author Response

Dear reviewer! Thanks a lot for your kind recommendations! It helps us to improve manuscript.

A new version of manuscript, modified according to another reviewer comments was uploaded.